# Modeling for Hysteresis Contact Behavior of Bolted Joint Interfaces with Memory Effect Penalty Constitution

Di Yuan [ID], Dong Wang [ID] and Qiang Wan *

Institute of Systems Engineering, China Academy of Engineering Physics, Mianyang 621999, China; yuandi128@outlook.com (D.Y.); king_east@sina.cn (D.W.)
* Correspondence: wanzhenyu20220607@126.com

**Abstract:** A novel penalty contact constitution was developed to replicate the hysteresis memory effect observed in contact interfaces. On this basis, a refined finite element analysis (FEA) was performed to study the stick–slip friction contact behavior of bolted joint interfaces. The analysis was validated by comparing it with the experimental hysteresis loops in the literature. The simulated hysteresis loops were subsequently used to identify four parameters of the Iwan model. Additionally, the effects of bolt clamping, friction coefficient, and excitation amplitude were individually examined. It was found that the deterioration in bolt clamping performance resulted in a decrease in both the equivalent joint stiffness and energy dissipation. Similarly, the reduction in the friction coefficient yielded a comparable impact. Furthermore, the identified model parameters of critical stick–slip force and displacement exhibited a quasi-linear relationship to the bolt preload and friction coefficient.

**Keywords:** bolted joints; stick–slip friction; memory effect penalty; hysteresis loop; Iwan model





## 1. Introduction

Bolted joints are widely used to transfer load and energy of engineering equipment structures [1]. Forced by a vibrational load, the stick–slip friction contact behavior of joint interfaces can cause non-resource energy dissipation and lead to a reduction in joint stiffness. This phenomenon significantly affects the complex nonlinear dynamic characteristics of assembled structures [2–4]. Modeling joint interfaces poses a persistent scientific challenge, as it requires estimating various factors such as asperity interaction, including elastic–plastic deformation, friction, lubrication, and wear [5,6].

In the early research of bolted joints, equivalent linearization methods such as the simplified spring-damping element [7], virtual material layer [8,9], and zero-thickness element [10,11] were primarily employed to address the phenomena of stick–slip, contact separation, and plastic deformation. Testing, modeling, parameter identification, and solution methods were employed in Refs. [12–14]. Notably, Nassar et al. [15] introduced a quartic pressure distribution between connectors and proposed a linear equivalent expression for the effective area, which influences the stiffness of bolted joints. Sethuraman et al. [16] assumed isotropic and uniformly distributed joint materials, employing linear elastic axisymmetric finite element analysis to evaluate member stiffness. Fu et al. [17] developed a mathematical theoretical model to establish a relationship between normal stiffness and damping of joint interfaces. They extracted joint parameters from experimental data and conducted qualitative analysis to understand the underlying mechanism. However, the linearized equivalent approaches ignored the critical physical information of bolted joint interfaces and indicated an inability to reproduce the resultant nonlinear characteristics [18]. Consequently, model parameters obtained under a specific experimental condition cannot be directly extrapolated to other load scenarios.

To overcome the limitations of equivalent linearization methods, numerous nonlinear joint models have been developed through rigorous experimental, analytical, and numerical

efforts by researchers from diverse disciplines [19–21]. These models are intentionally developed to describe the induced nonlinear dynamic behavior of bolted joints, with their expressions and parameters tailored to replicate the specific nonlinear dynamics of joint interfaces. Notable examples include the Iwan model [22,23], the Bouc–Wen model [24], the LuGre model [25], the Valanis model [26], etc. Among them, the Iwan model and its improved versions have been widely used to characterize the hysteresis nonlinearity induced by the stick–slip friction contact behavior of joint interfaces, since the model expressions and parameters indicated a good performance in describing the relatively transparent physical significance. The Iwan model consists of multiple Jenkins elements arranged in parallel or series, and has been extensively used to model the stick–slip friction contact behavior while the residual stiffness of a macro-slip regime was ignored. To rectify this deficiency, improved Iwan models with four parameters and six parameters were separately developed by Wang [27] and Li [28] to model the influence of residual stiffness, where an additional spring was implemented with the Iwan model in parallel.

Additionally, the finite element method has been employed to investigate the stick–slip contact friction behavior of bolted joint interfaces. Various contact constitution approaches have been utilized to simulate the friction contact behavior of the bolted joints, i.e., bonded constraint [29], the standard penalty method [30–32] and the Lagrange multiplier method [33]. However, these approaches mentioned above overlook the influence of micro-scale memory effect. Consequently, a significant difference between the simulated hysteresis loops and experimental observations was observed, particularly for the evolutionary process from the micro-stick to the macro-slip regime [34]. Therefore, it is crucial to develop a memory effect penalty model to effectively describe the stick–slip friction contact behavior of bolted joint interfaces and enhance the understanding of nonlinear mechanisms within these interfaces.

Moreover, forced by a long-time vibration load, the cumulative effects of wear and slip result in decreased clamping performance of bolted joint interfaces [35,36], known as bolt loosening [37]. The loosening phenomenon has a significant impact on the resultant nonlinear dynamic characteristics [38,39]. Therefore, to develop a nonlinear dynamic model of bolted joint interfaces effectively, it is essential to study the influence of bolt preload on the stick–slip friction contact behavior of joint interfaces and examine the key factors influencing hysteresis nonlinearity.

In the current study, a memory effect penalty contact constitution was developed to model the hysteresis nonlinearity induced by the stick–slip friction contact behavior of joint interfaces in Section 2. On this basis, a refined finite element of a bolted joint structure was conducted to investigate the hysteresis nonlinearity and verified by comparing the outcome with the literature results in Section 3. The hysteresis nonlinearity was used to identify four parameters of the Iwan model, and the effects of clamping performance, friction coefficient, and excitation amplitude were also investigated. Conclusions were drawn in Section 5.

## 2. Contact Model for Joint Interfaces

### 2.1. Standard Penalty Contact Method

The penalty contact method is commonly used to solve nonlinear contact problems [40–42]. In the separation regime shown in Figure 1a, the contact between surfaces is absent, where $\Gamma_t$ and $\Gamma_u$ represent traction and displacement components, respectively. The potential energy functional is given as

$$\prod(U) = \frac{1}{2}U^T K U - U^T F \tag{1}$$

where $U$ denotes the node displacement in the normal direction, $K$ is the stiffness matrix, and $F$ represents the external excitation forces.

As for the contact regime depicted in Figure 1b, a penalty potential energy term is incorporated into the potential energy functional, and that is

$$\overset{*}{\prod}(\boldsymbol{U}) = \frac{1}{2}\boldsymbol{U}^T\boldsymbol{K}\boldsymbol{U} - \boldsymbol{U}^T\boldsymbol{F} + \frac{1}{2}\boldsymbol{P}^T\boldsymbol{E}_p\boldsymbol{P} \tag{2}$$

where $\boldsymbol{P}$ denotes the penetration distance, which varies as a function of $\boldsymbol{U}$. $\boldsymbol{E}_p$ represents the penalty factor.

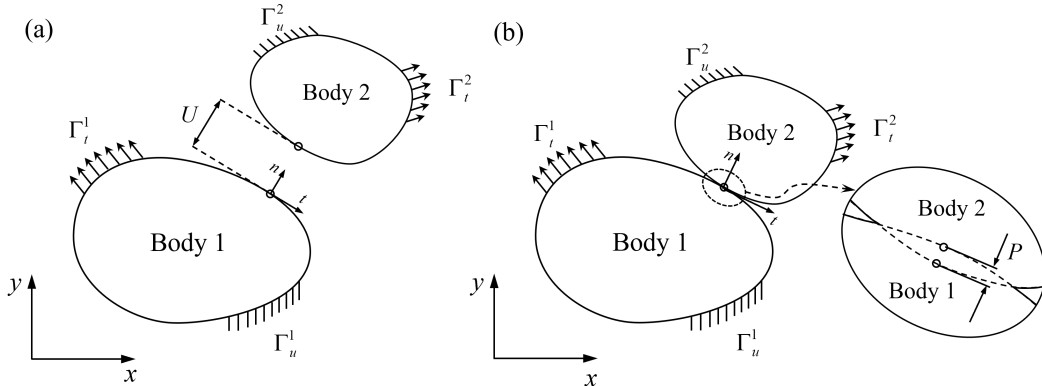

**Figure 1.** Relationship of two bodies: (**a**) Separation and (**b**) Contact.

The fundamental principle of the penalty method is to transform constrained problems into unconstrained ones by introducing a penalty function [43]. Using an iteration method to analyze $\boldsymbol{U}$, the governing equation can be formulated as

$$(\boldsymbol{K} + \boldsymbol{K}_p)\boldsymbol{U} = \boldsymbol{F} - \boldsymbol{F}_p \tag{3}$$

where $\boldsymbol{K}_p$ represents the penalty stiffness matrix, $\boldsymbol{K}_p = \left(\frac{\partial \boldsymbol{P}}{\partial \boldsymbol{U}}\right)^T \boldsymbol{E}_p \frac{\partial \boldsymbol{P}}{\partial \boldsymbol{U}}$. $\boldsymbol{F}_p$ denotes the correponding force matrix, $\boldsymbol{F}_p = \left(\frac{\partial \boldsymbol{P}}{\partial \boldsymbol{U}}\right)^T \boldsymbol{E}_p \boldsymbol{P}_0$.

It is observed that no relative interaction exists for separation nodes of the contact interfaces. Forced by an excitation load, the separation nodes undergo a gradual transformation into a contact regime characterized by a decreasing relative distance. To prevent node intrusion, a rigid spring is incorporated between them.

For the two-dimensional contact problem, the stress–strain relationship of the contact surfaces is described using the ideal Coulomb friction, as illustrated in Figure 2. The critical tangential slip stress is defined as

$$\tau_{\text{cr}} = \mu p \tag{4}$$

where $\mu$ represents the friction coefficient and $p$ denotes the normal contact pressure. If the tangential stress $\tau < \tau_{\text{cr}}$, there is no sliding movement, and the contact area remains in a stick contact state. Conversely, a relative sliding transpires, and the contact area exhibits a slip contact regime.

Referring to Equation (2), the elastic potential energy of the spring is utilized as the penalty term incorporated into the energy equation. Considering the constraint condition, a higher penalty factor can yield a more accurate approximation of the contact problem solution but may result in an ill-conditioned equation.

Therefore, the penalty factor is adaptive, chosen to ensure the convergence efficiency and calculation precision of simulating the friction contact behavior of joint interfaces. Especially when the penalty factor trends to infinity, the standard penalty method is degraded into a Lagrange one, labeled by the blue solid line in Figure 2.

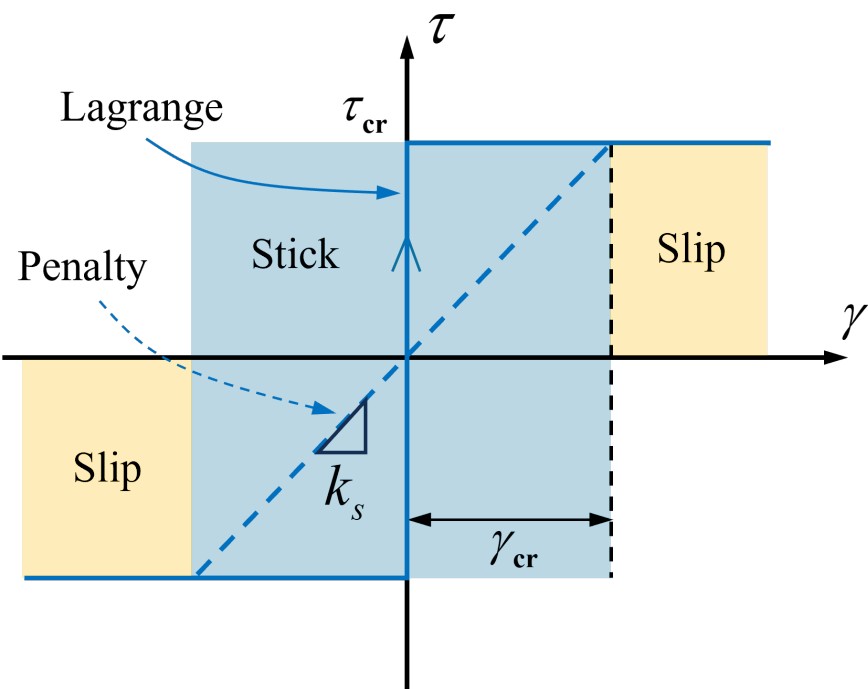

**Figure 2.** Stress–strain curve of Coulomb friction behavior.

Illustrated by the blue area in Figure 2, an adjusted high-stiffness linear spring is introduced to avert non-convergence stemming from abrupt transitions between stick and slip. This adjustment enables a smooth shift in the contact state, referred to as the "elastic slip" state or the "stick" state. The standard penalty method allows for a certain degree of compromise on accuracy to enhance computational efficiency. In the case of stick contact, the relationship between tangential stress and tangential relative deformation is defined as

$$\tau = k_s \gamma \tag{5}$$

where $k_s$ represents the stiffness of the rigid spring. $\gamma$ denotes the tangential relative deformation. The value of $k_s$ is determined by

$$k_s = \frac{\tau_{\text{cr}}}{\gamma_{\text{cr}}} \tag{6}$$

According to Equations (4)–(6), the value of $k_s$ is determined by normal contact pressure $p$. During the transition from the stick stage to the slip stage, the tangential stress and relative deformation exhibit a linear relationship with a slope ratio of stick stiffness. As for the macro-slip stage, the tangential stress is given by

$$\tau = \frac{\gamma}{|\gamma|} \mu p \tag{7}$$

The standard penalty method is utilized to model the contact interaction of joint interfaces. The relationship between the tangential stress $\tau$ and relative deformation $\gamma$ is given by

$$\tau(\gamma) = \begin{cases} k_s \gamma & \gamma < \gamma_{\text{cr}} \\ \mu p & \gamma \geq \gamma_{\text{cr}} \end{cases} \tag{8}$$

where $\gamma_{\text{cr}}$ denotes the critical deformation for stick–slip transformation.

Moreover, based on Equation (8) and Figure 2, it can be observed that the standard penalty method solely alters the direction during the reloading and unloading processes. Only a consistent relationship between the tangential stress and relative deformation can be considered.

### 2.2. Memory Effect Penalty Constitution

To capture the stick–slip friction contact behavior of joint interfaces, the standard penalty constitution requires an extremely tiny scattered element to describe the friction contact effects. It limits the contact regime to either stick or slip, following the dry friction principle. However, it is difficult to model the evolutionary transition from micro-stick to macro-slip regimes accurately. In this section, an improved penalty contact constitution is developed to model the stick–slip evolution effects and hysteresis memory effects.

The namely smooth surface is rough at a micro-scale view and covered with a multitude of asperities with varying shapes and sizes. Modeling for the stick–slip friction contact behavior involves the estimation of asperity integration, including the normal deformation and tangential friction properties. Corresponding to the dry friction constitution of a scattered element, the contact behavior of an individual asperity is studied to investigate the stick–slip friction behavior as shown in Figure 3.

As for the contact behavior of two spherical asperities subjected to a normal preload, a contact area with radius $R$ is established. Additionally, forced by a tangential load, the non-uniform distribution of contact pressure causes the edge of the contact region to experience initial tangential slip. The region without slip is termed the "stick" region, characterized by radius $r$ while the slip region forms a ring with width $(R - r)$. As the tangential load intensifies, the slip region gradually expands and contracts toward the center, eventually resulting in macro-slip across the entire contact region.

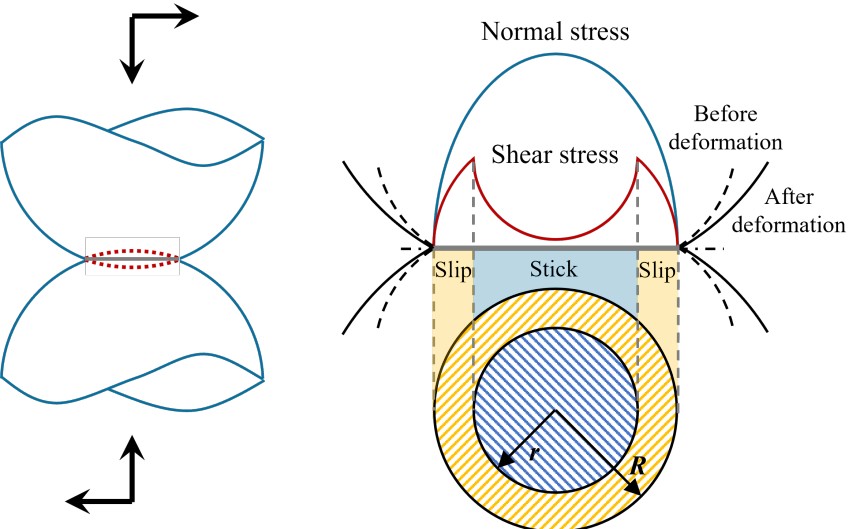

**Figure 3.** Schematic diagram of asperity contact.

By combining the Hertz's contact theory with Coulomb's friction law, the Mindlin solution [44] is used to model the tangential stick–slip contact interaction. The relationship between tangential stress and relative deformation is given by

$$F(\gamma) = \begin{cases} \mu p \left[ 1 - \left( 1 - \frac{\gamma}{\gamma_{cr}} \right)^{\chi} \right] & \gamma < \gamma_{cr} \\ \mu p & \gamma \geq \gamma_{cr} \end{cases} \tag{9}$$

where $\mu$ represents the friction coefficient and $p$ denotes the normal pressure. $\gamma_{cr}$ is the critical stick–slip deformation at the initiation of macro-slip contact. $\chi$ denotes the power exponent that represents the nonlinear degree of tangential stress and relative deformation, which depends on the roughness of the surface.

Both Equations, (8) and (9), can be used to model the tangential contact behavior of stick and slip regimes. However, a distinguishing factor in Equation (9) is substituted with a nonlinear equation represented as a power function, while a linear equation of the stick stage is defined in the standard penalty method in Equation (8).

Forced by a periodic load, the cyclic loading process is categorized into distinct reloading and unloading processes. In accordance with the Masing's principle, the tangential stress for each loading process can be defined as

$$F_{\text{rel}}(\gamma) = -F_0 + 2F\left(\frac{\gamma_{\text{cr}}+\gamma}{2}\right) \quad \dot{\gamma} \geq 0 \tag{10}$$

$$F_{\text{unl}}(\gamma) = F_0 - 2F\left(\frac{\gamma_{\text{cr}}-\gamma}{2}\right) \quad \dot{\gamma} < 0 \tag{11}$$

where $F_0$ represents the amplitude of tangential stress and depends on relative deformation amplitude. And $\dot{\gamma}$ is the incremental relative slip deformation. Subscripts "rel" and "unl" represent the reloading and unloading processes, respectively.

By integrating Equation (9) into Equations (10) and (11), the tangential stress of the cyclic loading process is defined as

$$\begin{cases} F_{\text{rel}}(\gamma) = -\mu p + 2\mu p\left[1 - \left(1 - \frac{\gamma_{\text{cr}}+\gamma}{2\gamma_{\text{cr}}}\right)^{\chi}\right] & \dot{\gamma} \geq 0 \\ F_{\text{unl}}(\gamma) = \mu p - 2\mu p\left[1 - \left(1 - \frac{\gamma_{\text{cr}}-\gamma}{2\gamma_{\text{cr}}}\right)^{\chi}\right] & \dot{\gamma} < 0 \end{cases} \tag{12}$$

By amalgamating Equations (8), (9) and (12), the complete loading process can be comprehensively expressed, encompassing the initial loading as well as the cyclic loading stages.

$$F(\gamma) = \begin{cases} \mu p\left[1 - \left(1 - \frac{\gamma}{\gamma_{\text{cr}}}\right)^{\chi}\right] & \gamma < \gamma_{\text{cr}} & \text{Initial loading} \\ \mu p - 2\mu p\left[1 - \left(1 - \frac{\gamma_{\text{cr}}-\gamma}{2\gamma_{\text{cr}}}\right)^{\chi}\right] & \dot{\gamma} < 0 \,\&\, \gamma < \gamma_{\text{cr}} & \text{Unloading} \\ -\mu p + 2\mu p\left[1 - \left(1 - \frac{\gamma_{\text{cr}}+\gamma}{2\gamma_{\text{cr}}}\right)^{\chi}\right] & \dot{\gamma} \geq 0 \,\&\, \gamma < \gamma_{\text{cr}} & \text{Reloading} \\ \mu p & \gamma \geq \gamma_{\text{cr}} & \text{Slip} \end{cases} \tag{13}$$

Equation (13) epitomizes the two-dimensional manifestation of the proposed memory effect penalty constitution. The main difference between the proposed constitution and the standard penalty method lies in the transformation in the stick region as shown in Figure 4a. The comparison results show that the proposed constitution indicates a more accurate representation of the stick–slip mechanism characteristic of the contact interfaces.

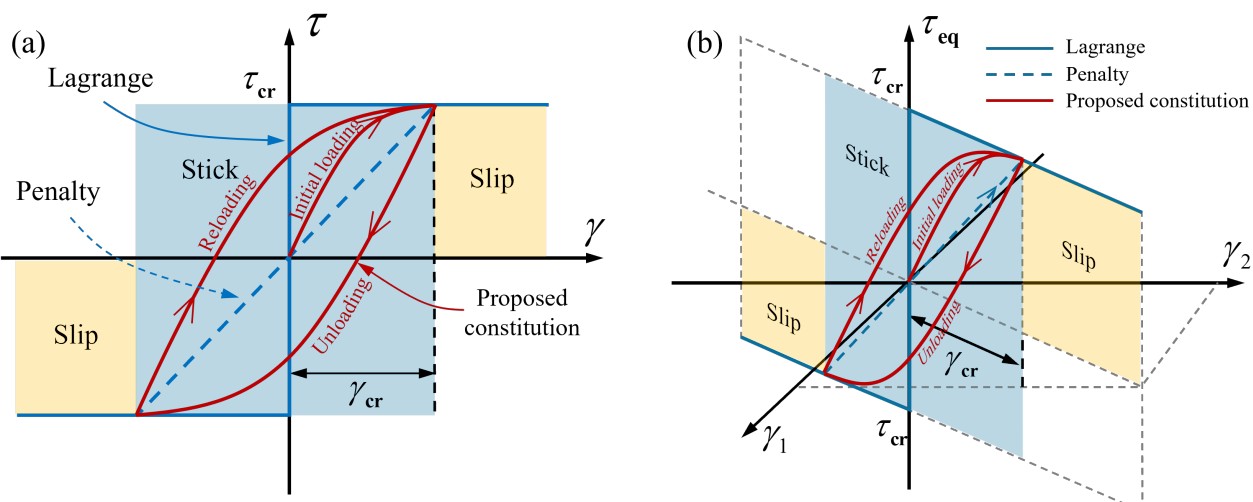

**Figure 4.** Stress–strain curve of memory effect penalty constitution: (**a**) 2D model and (**b**) 3D model.

For three-dimensional contact problems, the complexity increases for two degrees of freedom ($\gamma_1$ and $\gamma_2$) for deformation within the contact plane. It is assumed that the frictional characteristics in both directions are isotropic. Similar to the two-dimensional contact problem, the tangential friction occurs between the two surfaces in the plane formed

by $\gamma_1$ and $\gamma_2$. When the tangential stress satisfies $\tau_{eq} < \tau_{cr}$, the two surfaces are in a stick contact state. When the tangential stress exceeds the critical tangential stress with $\tau_{eq} \geq \tau_{cr}$, the sliding stage is initiated, where $\tau_{eq}$ represents the equivalent tangential stress and is defined as

$$\tau_{eq} = \sqrt{\tau_1^2 + \tau_2^2} \tag{14}$$

where $\tau_1$ and $\tau_2$ denote the tangential stresses along the two across tangential directions.

The expression for equivalent relative deformation is given by

$$\gamma_{eq} = \sqrt{\gamma_1^2 + \gamma_2^2} \tag{15}$$

By considering the isotropic of friction nature, the critical tangential stress is expressed as

$$\tau_i = \mu p \left[ 1 - \left( 1 - \frac{\gamma_i}{\gamma_{cr}} \right)^\chi \right], \text{ for } i = 1, 2 \tag{16}$$

The incremental form of Equation (16) during the stick contact regime is defined as

$$d\tau_i = \mu dp \left[ 1 - \left( 1 - \frac{\gamma_i}{\gamma_{cr}} \right)^\chi \right] + \frac{\mu p \chi}{\gamma_{cr}} \left( 1 - \frac{\gamma_i}{\gamma_{cr}} \right)^{\chi-1} d\gamma_i, \text{ for } i = 1, 2 \tag{17}$$

Equation (17) is used for the contact subroutine mentioned later. By combining Equations (14)–(16), the tangential stress of monolithic loading process is defined as

$$F(\gamma_{eq}) = \begin{cases} \mu p \left[ 1 - \left( 1 - \frac{\gamma_{eq}}{\gamma_{cr}} \right)^\chi \right] & \gamma_{eq} < \gamma_{cr} \\ \mu p & \gamma_{eq} \geq \gamma_{cr} \end{cases} \tag{18}$$

The tangential stress of the oscillatory loading process is given by

$$F_{rel}(\gamma_{eq}) = -F_0 + 2F\left( \frac{\gamma_{cr} + \gamma_{eq}}{2} \right) \quad \dot{\gamma}_{eq} \geq 0 \tag{19}$$

$$F_{unl}(\gamma_{eq}) = F_0 - 2F\left( \frac{\gamma_{cr} - \gamma_{eq}}{2} \right) \quad \dot{\gamma}_{eq} < 0 \tag{20}$$

By substituting Equation (18) into Equations (19) and (20), the relationship between tangential stress and relative deformation is given by

$$\begin{cases} F_{rel}(\gamma_{eq}) = -\mu p + 2\mu p \left[ 1 - \left( 1 - \frac{\gamma_{cr} + \gamma_{eq}}{2\gamma_{cr}} \right)^\chi \right] & \dot{\gamma}_{eq} \geq 0 \\ F_{unl}(\gamma_{eq}) = \mu p - 2\mu p \left[ 1 - \left( 1 - \frac{\gamma_{cr} - \gamma_{eq}}{2\gamma_{cr}} \right)^\chi \right] & \dot{\gamma}_{eq} < 0 \end{cases} \tag{21}$$

The expressions for tangential stress and relative deformation throughout a complete loading cycle are provided as follows:

$$F(\gamma_{eq}) = \begin{cases} \mu p \left[ 1 - \left( 1 - \frac{\gamma_{eq}}{\gamma_{cr}} \right)^\chi \right] & \gamma_{eq} < \gamma_{cr} & \text{Initial loading} \\ \mu p - 2\mu p \left[ 1 - \left( 1 - \frac{\gamma_{cr} - \gamma_{eq}}{2\gamma_{cr}} \right)^\chi \right] & \dot{\gamma}_{eq} < 0 \& \gamma_{eq} < \gamma_{cr} & \text{Unloading} \\ -\mu p + 2\mu p \left[ 1 - \left( 1 - \frac{\gamma_{cr} + \gamma_{eq}}{2\gamma_{cr}} \right)^\chi \right] & \dot{\gamma}_{eq} \geq 0 \& \gamma_{eq} < \gamma_{cr} & \text{Reloading} \\ \mu p & \gamma_{eq} \geq \gamma_{cr} & \text{Slip} \end{cases} \tag{22}$$

As seen from Equation (22), a memory effect penalty constitution is derived from the contact theory of an individual asperity in this study. In the context of FEA for the contact problem, the iteration process is determined by estimating the contact state of each contact element. By incorporating the memory effect constitution into the finite element model to describe tangential contact behavior, each element of the contact surface can be regarded as an asperity model. This representation enhances the capability to capture micro-scale stick–slip contact mechanisms.

The main steps of contact iteration are summarized as follows:

1. Judgment is performed to determine the contact state and the loading phase.
2. Based on the loading phase, the equivalent tangential stress is calculated.
3. A stick–slip contact state is determined based on the equivalent tangential stress.
4. The contact deformation and pressure increments are calculated.

To apply the proposed contact constitution in practical scenarios, the Fortran programming language is utilized to compile the memory effect penalty constitution into a finite element user-defined subroutine for modeling tangential contact behavior in joint interfaces. In addition, the contact pressure on bolted joints is non-uniform. Compared with existing nonlinear contact models (e.g., the Iwan model), this method does not require consideration of the pressure distribution characteristics on the contact interface. The prior calculation of contact pressure for each element during the determination of contact makes it easier to obtain the pressure distribution across the entire contact interface, thereby enhancing the method's versatility.

### 2.3. Comparison of Different Contact Methods

In order to highlight the performance of the proposed penalty constitution in capturing the micro-scale memory effect of stick–slip frictional contact problems, a comparative analysis between the proposed method, the standard penalty method, and the Lagrange multiplier method is conducted though a simple numerical example.

As shown in Figure 5, we consider a static frictional contact problem between two elastic bodies (a flat and a slider), in which the dimensions of the flat are 80 mm × 30 mm × 10 mm and the slider is a cube with a side length of 10 mm. The material used in the simulation is steel; its Young's modulus and Poisson's ratio are 200 GPa and 0.3, respectively. The friction coefficient is consistently set at 0.43.

The calculation process comprises two analysis steps. Firstly, a static simulation is performed to model the compressive process by applying a constant normal pressure to the upper surface of the slider, while the lower surface of the flat is fixed. Then, a displacement load is applied to the right side of the slider in the *x*-direction to generate slip behavior. The expression for the displacement excitation is described as follows:

$$A = A_0 \sin(2\pi t) \tag{23}$$

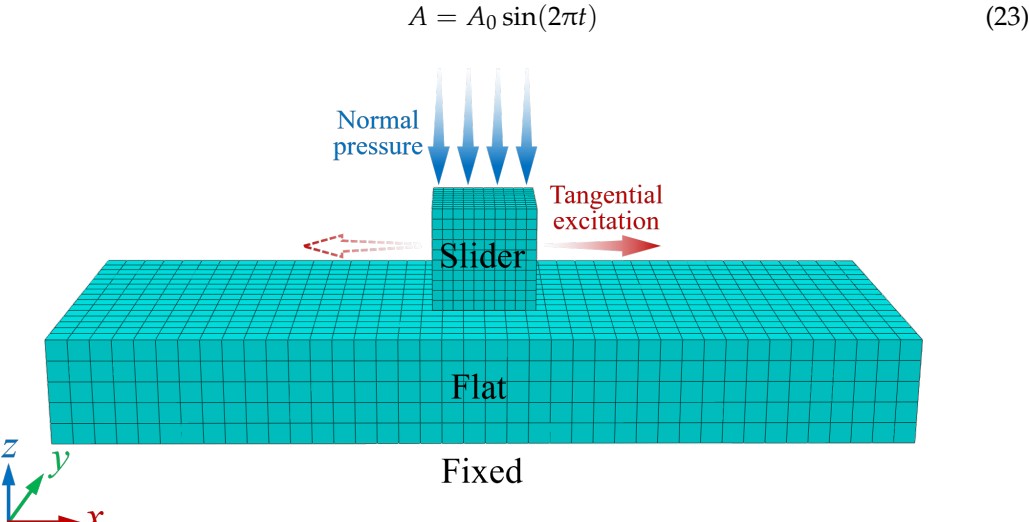

**Figure 5.** Finite element model of lap flat.

The simulated hysteresis loops considering three contact methods are shown in Figure 6, which are constructed by establishing the relationship between the tangential relative displacement and the tangential force of the contact surface. It can be observed that the tangential forces obtained by the three methods are identical during the macro-slip phase. The main difference lies in the tangential stiffness during the stick-to-micro-slip

phase. The hysteresis loop calculated by the standard penalty method closely resembles a parallelogram shape. Furthermore, there is an approximate linear relationship between the tangential relative displacement and tangential force during the stick and micro-slip phase. This phenomenon can be attributed to the "elastic slip" effect as mentioned above. In contrast, the hysteresis loop obtained by the Lagrange multiplier method assumes an almost rectangular form, which is characterized by only two stages: stick and macro-slip. Moreover, the tangential stiffness during the loading phase is significantly higher than that of the standard penalty method. However, when using the memory effect penalty method, the simulated hysteresis loop is distinct from the other two results. Specifically, the whole evolution curve is divided into three stages: stick, micro-slip, and macro-slip. The transitions between these stages manifest more smoothly, enabling a more accurate simulation of the nonlinear stiffness softening process and bringing the simulated results closer to the theoretical predictions.

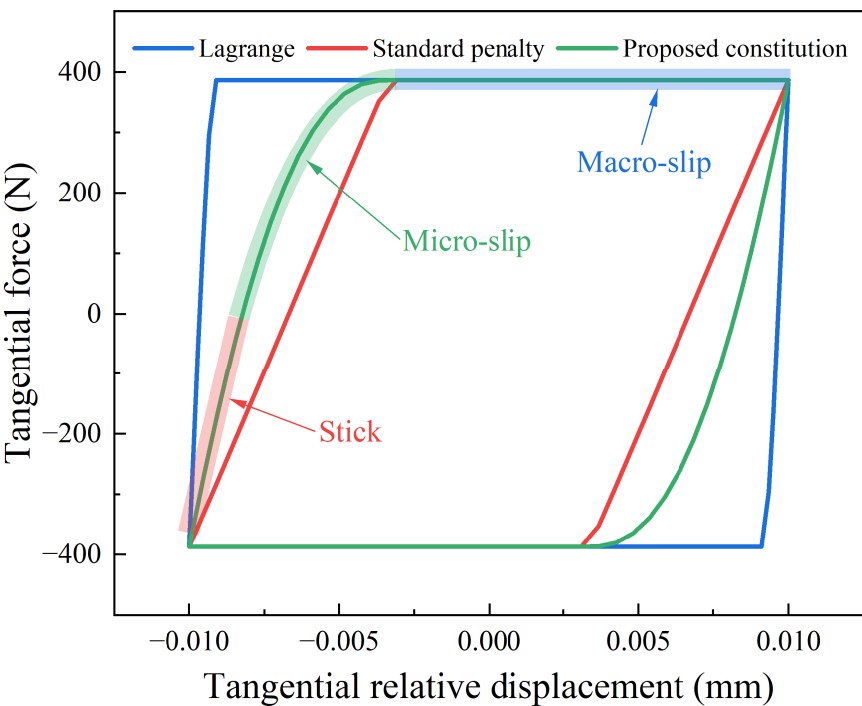

**Figure 6.** Simulated hysteresis loop with three different contact algorithms.

## 3. Refined FEA for Bolted Joint

### 3.1. Simulation Setup

On the basis of the proposed memory effect contact constitution, a refined finite element model of a bolted joint is constructed and validated by comparison with the experimental results in Ref. [34]. The schematic diagram illustrating the experimental setup for measuring the hysteresis loops of the lap-type bolted joint is shown in Figure 7a. The type of bolt is M6, carbon steel. The nominal contact region of the specimen consists of a 20 mm × 20 mm square with a 7 mm diameter through-hole. As shown in Figure 7b, a force washer is introduced to the bolt head to faithfully replicate the experimental conditions, and its material properties are the same as those of the bolt.

Due to the potential for nut rotation during the loading process, which may result in reduced bolt clamping performance, thereby influencing the overall mechanical characteristics of the bolted joints, the thread was retained and its modeling methodology drawn from Ref. [45] as shown in Figure 7c. The entire FE model contains 138,730 nodes and 124,748 elements. Five distinct contact pairs are established: bolt head-force washer, force washer-fixed specimen, fixed specimen-moving specimen, moving specimen nut and bolt thread nut thread from top to bottom. For computational efficiency, the contact pair of the

"fixed specimen-moving specimen" adopts the memory effect penalty constitution, while the remaining contact pairs use the standard penalty method. Due to the ample space in the through-hole and the relatively small amplitude of the external excitation, the screw-through-hole contact is disregarded to enhance computational efficiency. The contact area is marked by orange lines in Figure 7b.

In the simulation, the left surface of the fixed specimen is fully constrained ($u_1 = u_2 = u_3 = 0$). On the right surface of the moving specimen, a sinusoidal displacement excitation load is applied along the $x$-axis, and the degrees of freedom for the $y$ and $z$ axes are constrained ($u_1 = A \sin(2\pi t)$, $u_2 = u_3 = 0$), where $A$ represents the excitation amplitude.

Figure 8 presents the typical hysteresis loop for a bolted joint subjected to a tangential cyclic load. The hysteresis loop can be segmented into the reloading phase and the unloading phase, symmetrically oriented around the origin. Furthermore, the unidirectional loading process can be further divided into three stages: stick, micro-slip, and macro-slip. The tangential stiffness of the stick stage is referred to as stick stiffness ($k_t$), while that of the macro-slip stage is called residual stiffness ($k_\alpha$).

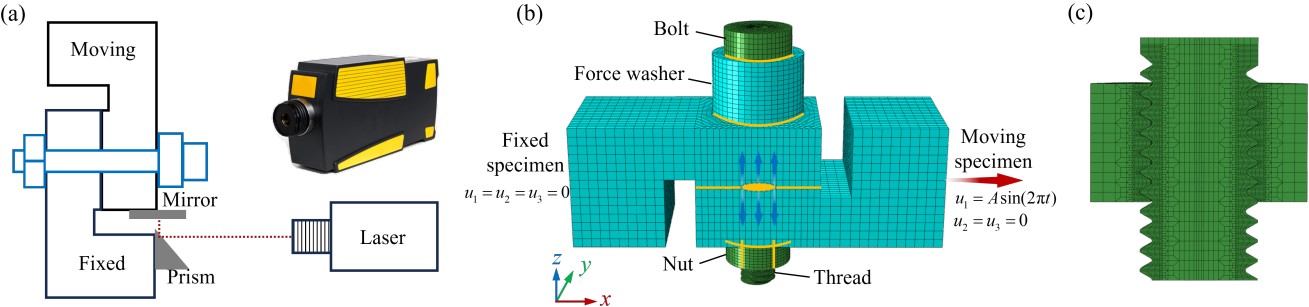

**Figure 7.** Schematic diagram of bolted joint: (**a**) Experimental setup [45], (**b**) Refined finite element model, and (**c**) Mesh of thread.

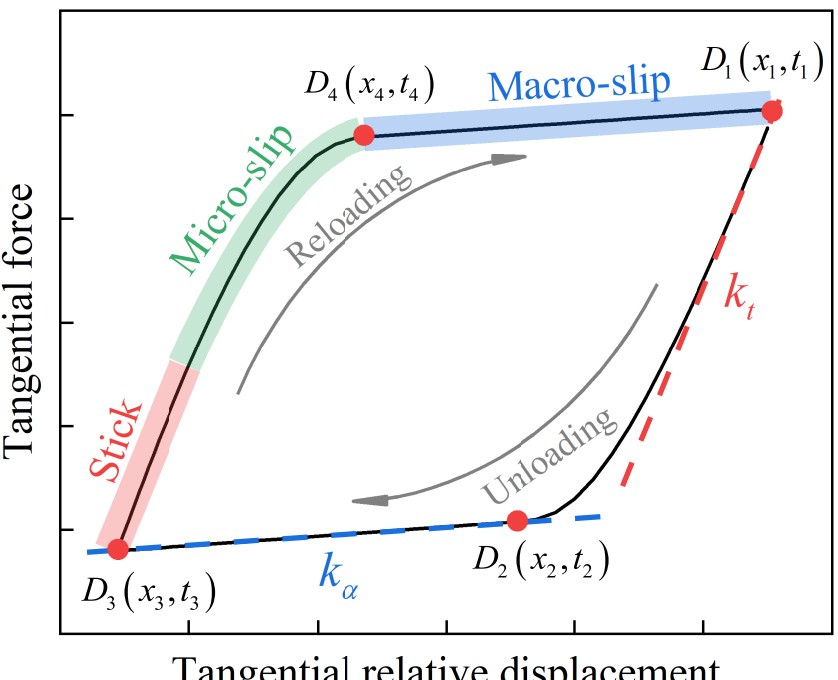

**Figure 8.** Typical hysteresis loop of bolted joint.

*3.2. Model Validation*

In the simulation, the elastic slip distance for both the standard penalty method and the memory effect penalty constitution are set to 0.4%, while the power exponent $\chi$ of the memory effect penalty constitution is set to 2.5. Three different clamping forces (327 N, 592 N, and 902 N) are considered, and the corresponding friction coefficients (0.29, 0.37 and 0.42) are derived from the respective hysteresis loops. The comparative results are presented in Figure 9.

The hysteresis loops of the proposed method, as shown in Figure 9, exhibit good agreement with the experimental results under three different preloading conditions. In contrast, the hysteresis loops simulated by the Lagrange multiplier method (blue line) and the standard penalty method (green line) demonstrate satisfactory fitting performance during the macro-slip stage. However, these simulations maintain a predominantly linear tangential stiffness during the stick and micro-slip stages, failing to capture the nonlinear stiffness softening characteristics of the bolted joint. The comparative analysis attests the capability and validity of the proposed penalty constitution in capturing the micro-scale stick–slip behavior at the bolted joint interface.

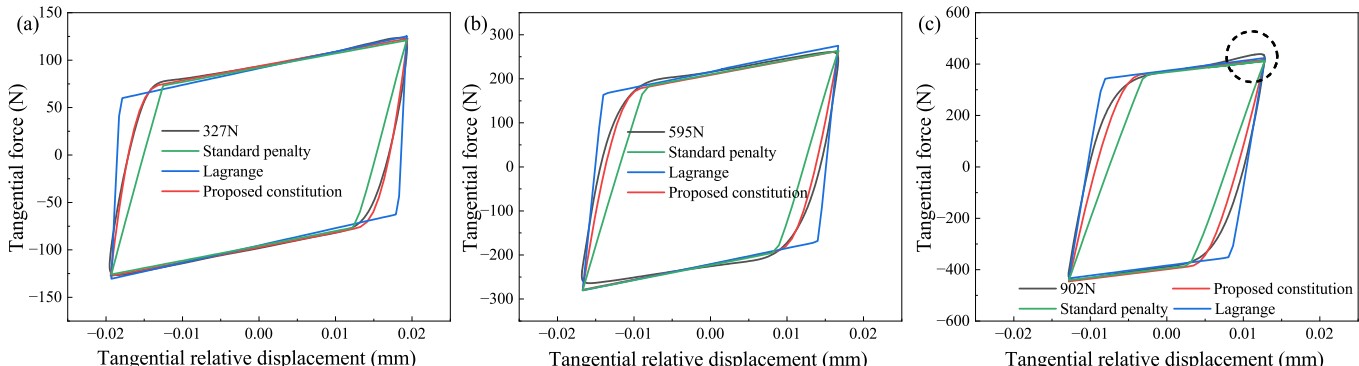

**Figure 9.** Comparisons between finite element results and experimental results [34]: (**a**) Hysteresis loop with a preload of 327 N, (**b**) Hysteresis loop with a preload of 595 N, and (**c**) Hysteresis loop with a preload of 902 N.

To further validate the performance of the proposed memory effect penalty constitution, the energy dissipation results obtained from the three contact methods are compared with the experimental results. The area enclosed by the hysteresis loop corresponds to the total energy dissipation during a cyclic loading period [46]. The areas of the hysteresis loops shown in Figure 9 are calculated through mathematical integration. Subsequently, an error analysis is conducted to compare the numerical results with the experimental data, as shown in Figure 10; it can be observed that the memory effect penalty constitution yields a lower average error compared to the other two methods. It is worth noting that in the experimental results under the 902 N condition, in Figure 9c, a phenomenon of hysteresis loop uplift was observed. According to the literature [34], this phenomenon is mainly attributed to assembly uncertainties, resulting in a minor increase in the hysteresis loop area and a subsequent rise in the associated errors.

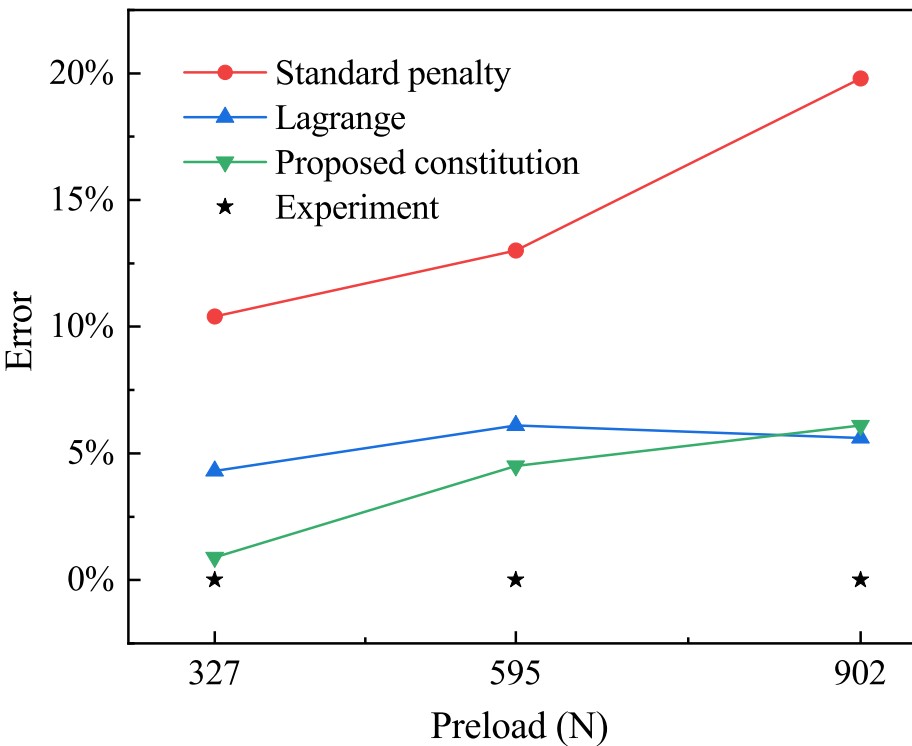

**Figure 10.** Comparison of energy dissipation in hysteresis loops.

### 3.3. Parameter Investigation

The preceding section demonstrates that the proposed penalty constitution effectively captures the micro-scale stick–slip behavior at the interface of the bolted joint. Based on the method, here, the dependencies of hysteresis characteristics of bolted joint on various parameters, such as bolt preload, friction coefficient, and external excitation amplitude, are investigated. Specifically, five parameters are designed for each working condition, and the detailed parameter setups are listed in Table 1.

**Table 1.** Finite element simulation parameters.

| $\mu = 0.3$ $A = 0.02$ mm | | Preload = 900 N $A = 0.02$ mm | | Preload = 900 N $\mu = 0.3$ | |
|---|---|---|---|---|---|
| Preload/N | 300 500 700 900 1100 | Friction coefficient ($\mu$) | 0.1 0.2 0.3 0.4 0.5 | External excitation amplitude ($A$)/mm | 0.005 0.010 0.015 0.020 0.025 |

Figure 11 illustrates the relationships between hysteresis characteristics of bolted joint and different parameters computed by FEA mentioned before. From Figure 11a and Figure 11c, it is evident that reducing the bolt preload and friction coefficient leads to a gradual decrease in the stick stiffness of the bolted joint. However, as seen in Figure 11e, the external excitation amplitudes have no influence on the stick stiffness. These results illustrate that the stick stiffness relies on both preload and friction coefficients, while it is independent of the excitation amplitude. Furthermore, the energy dissipation results under different parameter conditions are also displayed in Figure 11b,d,f. It can be seen that the energy dissipation decreases linearly with the reduction in preload, friction coefficient, and excitation amplitude. Moreover, as shown in Figure 11e, the hysteresis loops maintain an almost parallel disposition during the stick-to-micro-slip phase and entirely coincide during the macro-slip phase. Namely, the increase in external excitation amplitude does

not affect the stick stiffness and residual stiffness but does impact energy dissipation. The validity of this result can be justified by Equation (22), which indicates that a decrease in normal force and friction coefficient lead to a reduction in tangential force, thereby resulting in a decrease in tangential stiffness. Meanwhile, external excitation amplitude has little effect on tangential stiffness as it is independent of tangential force. However, an increase in excitation amplitude implies an increase in external work performed, resulting in an increase in energy dissipation.

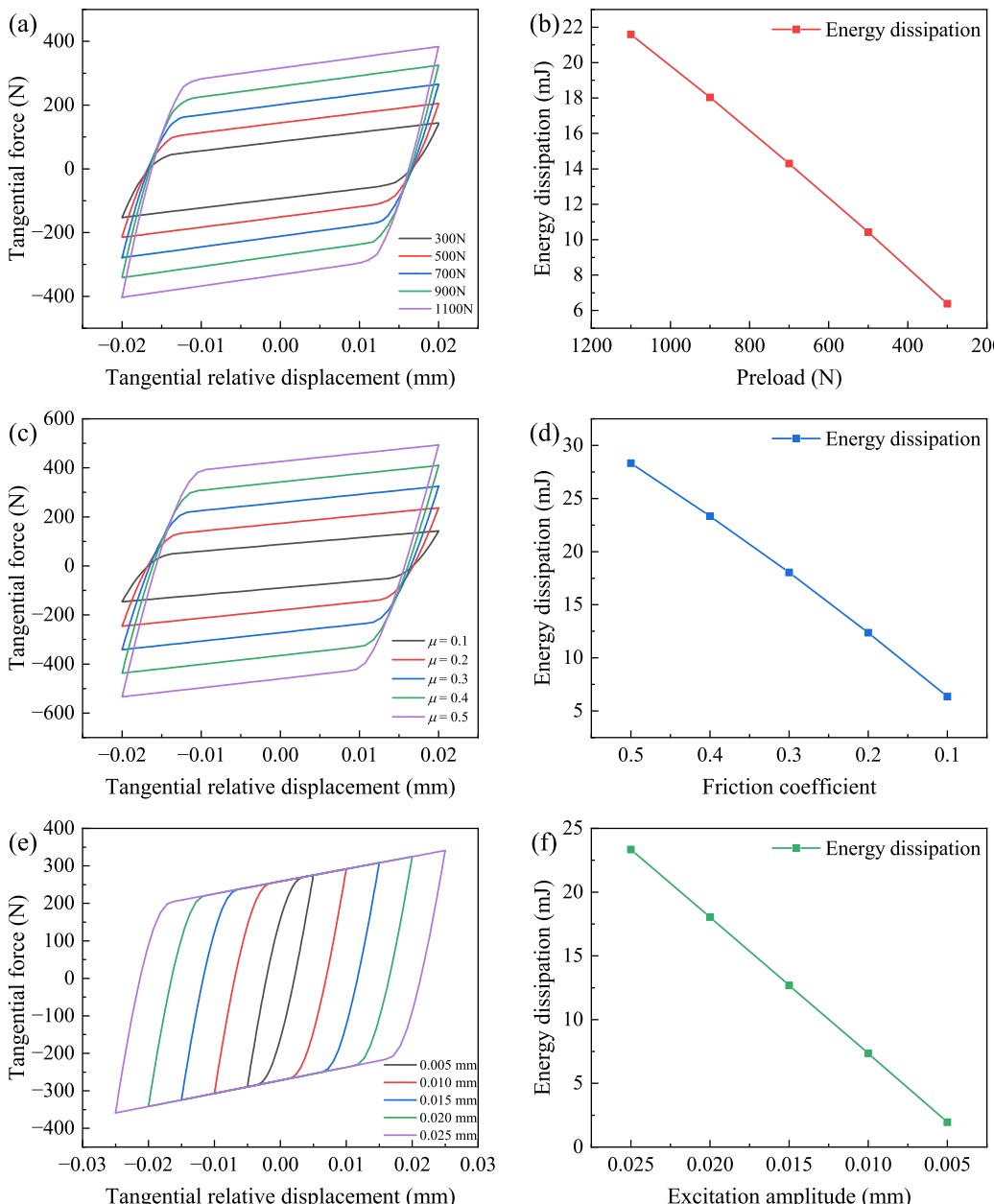

**Figure 11.** Hysteresis loops simulated by finite element: (**a**) Hysteresis loops for different preloads, (**b**) Energy dissipation values for different preloads, (**c**) Hysteresis loops for different friction coefficients, (**d**) Energy dissipation values for different friction coefficients, (**e**) Hysteresis loops for different external excitation amplitudes, and (**f**) Energy dissipation values for different external excitation amplitudes.

## 4. Modeling for Bolted Joint Interfaces

### 4.1. Reduced-Order Models (ROMs)

As presented in Section 1, the improved Iwan model can characterize the hysteresis nonlinearity induced by the stick–slip friction contact behavior of joint interfaces. In this section, the improved four-parameter Iwan model is utilized to identify the parameters from the finite element simulation results. The improved four-parameter Iwan model consists of four essential parameters $\{F_s, X_s, k_\alpha, \kappa\}$. Parameter $F_s$ denotes the critical tangential force transitioning from the micro-slip to the macro-slip, which is determined by the normal force and friction coefficient, and its value depends on the coordinate values of $D_2$ or $D_4$ in Figure 8. Parameter $X_s$ represents the critical stick–slip displacement, defining the essential relative displacement for the shift from the micro-slip to the macro-slip. The value of $X_s$ primarily depends on factors such as material properties, normal force, and friction coefficient. Specifically, $X_s$ is equivalent to half of the difference between the abscissae of $D_4$ and $D_3$ in Figure 8. Parameter $k_\alpha$ denotes the residual stiffness, which is predominantly contingent upon the material properties of the screw and the roughness of the contact surface. Parameter $\kappa$ represents a power exponent, reflecting the extent of nonlinearity in tangential stiffness during the stick-to-micro-slip stage. Its determination is predominantly influenced by the material properties and roughness of the contact surface. Once the values of these four parameters are determined, a definite hysteresis loop can be generated to fit the simulated or experimental results.

The simulated or experimental results generally comprise numerous scattered data points, which can be processed using a program to identify the coordinates of key points. Among these, the most easily discernible points are the maximum and minimum tangential displacement points, denoted as $D_1$ and $D_3$ in Figure 8, respectively. Once $D_1$ or $D_3$ are nearby, scattered points in the same direction can be selected and fitted to generate a line. The slope of this line represents residual stiffness $k_\alpha$. Subsequently, scatter points from the reloading or unloading phases are selected, and a linear relationship evaluation is performed on the line representing the residual stiffness. Within a tolerance range, the coordinates of the point with the maximum absolute displacement are chosen. This determination assists in establishing critical stick–slip displacement $X_s$, thereby enabling the derivation of critical stick–slip force $F_s$. If data from the reloading stage are selected, power exponent $\kappa$ can be solved using a double-sided logarithmic method, which can be represented by the following equation:

$$1 - \frac{T_{\max} + T - 2k_\alpha x}{2F_s} = \left(1 - \frac{x}{X_s}\right)^{\frac{5}{2} + \kappa} \tag{24}$$

Similarly, data from the unloading stage can also be utilized to address the problem. Thus far, the four parameters of the Iwan model can be identified; then, the study of the identified parameters can be performed systematically.

### 4.2. Results and Discussion

#### 4.2.1. Effect of Bolt Preload

The detailed identification data are obtained through parameter identification using simulated results under various preload conditions, as shown in Table 2. These data are then fitted to the simulation results, as illustrated by the dashed line in Figure 12a. However, due to the inherent asymmetry in reloading and unloading during the simulation process, there exists a symmetrical discrepancy between the four-parameter Iwan model and the simulated results. Even so, the associated error minimally affects the identification outcomes.

**Table 2.** Identification results of different bolt preloads.

| Preload (N) | Residual Stiffness (N/mm) | Critical Stick–Slip Force (N) | Critical Stick–Slip Displacement (mm) | Power Exponent |
|---|---|---|---|---|
| 300 | 2996.4 | 89.70 | 0.0032 | −0.644 |
| 500 | 3058.9 | 149.02 | 0.0036 | −0.611 |
| 700 | 3158.3 | 208.09 | 0.0039 | −0.755 |
| 900 | 3280.7 | 266.78 | 0.0041 | −0.842 |
| 1100 | 3351.4 | 325.38 | 0.0044 | −0.931 |

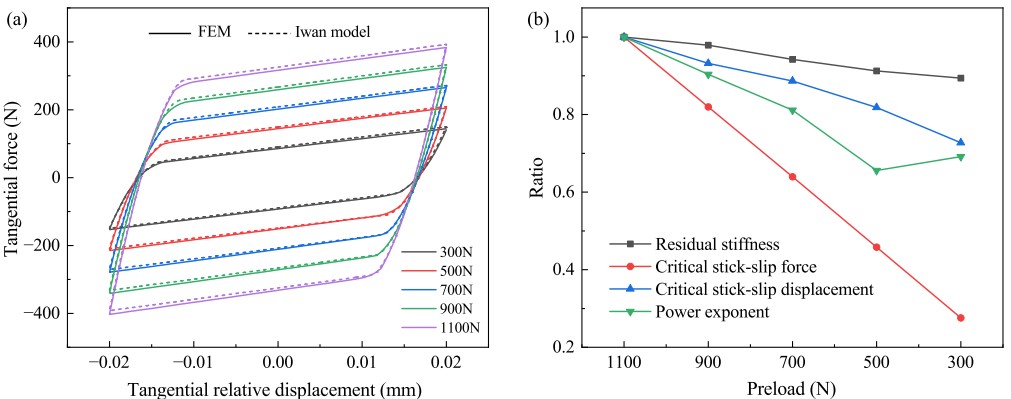

**Figure 12.** Modeling for bolted joint with clamping degradation: (**a**) Hysteresis nonlinearity and (**b**) Identified four parameters of Iwan model.

The results of identification in Table 2 are normalized, allowing for an exploration of the impact of bolt clamping performance degradation on hysteresis characteristics. As depicted in Figure 12b, the reduction in bolt preload exhibits minimal influence on residual stiffness. Specifically, an 800 N decrease in preload corresponds to an approximately 10% decrease in residual stiffness. The normalized critical stick–slip displacement and force exhibit a strong linear relationship with the decrease in bolt preload. Among these, preload exerts the most pronounced effect on critical stick–slip force. For instance, when bolt preload is diminished from 1100 N to 300 N, a preload loss of roughly 73% accompanies a reduction of about 72% in critical stick–slip force. This reveals an equivalent proportional decline in bolt preload and critical stick–slip force. Meanwhile, the critical stick–slip displacement experiences a reduction of approximately 27%. Moreover, the deterioration in bolt clamping performance also contributes to a decrease in the power exponent. This phenomenon implies that the nonlinear effect of the bolt joint interfaces diminishes as the transition occurs from the stick to the micro-slip stage.

### 4.2.2. Effect of Friction Coefficients

By varying the friction coefficients, simulations are performed to model different surface roughness conditions. The identified parameters are then obtained by fitting the finite element simulation results using the four-parameter Iwan model. The specific results are presented in Table 3, and the fitting performance is illustrated in Figure 13a.

**Table 3.** Identification results of different friction coefficients.

| Friction Coefficients | Residual Stiffness (N/mm) | Critical Stick–Slip Force (N) | Critical Stick–Slip Displacement (mm) | Power Exponent |
|---|---|---|---|---|
| 0.1 | 2753.1 | 89.03 | 0.0032 | −0.551 |
| 0.2 | 3137.3 | 150.98 | 0.0037 | −0.688 |
| 0.3 | 3280.7 | 219.05 | 0.0041 | −0.748 |
| 0.4 | 3338.4 | 307.33 | 0.0048 | −0.828 |
| 0.5 | 3365.2 | 393.15 | 0.0053 | −0.931 |

Similarly, the identification results detailed in Table 3 are normalized, as illustrated in Figure 13b. Notably, when the friction coefficient decreases from 0.5 to 0.1, the residual stiffness decreases by about 18%. This finding indicates that the influence of friction coefficient on residual stiffness surpasses that of bolt preload. Consistent with the results in Section 4.2.1, the normalized critical stick–slip force and displacement exhibit linear decrease in correspondence with diminishing friction coefficients. Specifically, with an 80% decrease in the friction coefficient, the critical stick–slip force decreases by 77%. This implies an equivalent proportional reduction between the friction coefficient and the critical stick–slip force. Furthermore, the decline in the friction coefficient impacts the power exponent more significantly than the reduction in bolt clamping performance. This alignment with common understanding stems from the fact that a decrease in the friction coefficient signifies a decline in surface roughness, thereby affecting the nonlinear influence of joint interfaces.

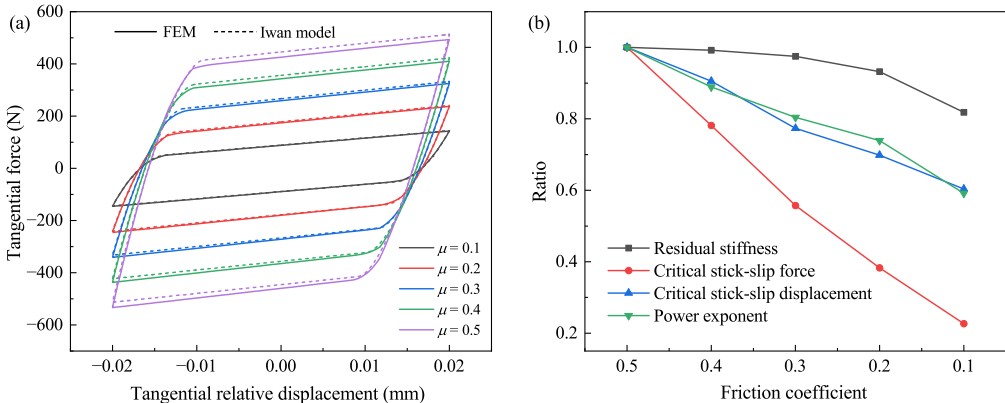

**Figure 13.** Modeling for bolted joint with different friction coefficient: (**a**) Hysteresis nonlinearity and (**b**) Identified four parameters of Iwan model.

### 4.2.3. Effect of External Excitation Amplitudes

This section aims to investigate the influence of external excitation amplitude on the hysteresis characteristics of bolted joints. The identified results, presented in Table 4, are obtained using the four-parameter Iwan model. The corresponding fitting curve is illustrated in Figure 14a. Following the parameter identification process, normalization procedures are performed to derive the parameter curves related to external excitation amplitudes. Detailed illustrations of these curves can be found in Figure 14b.

**Table 4.** Identification results of different external excitation amplitudes.

| Excitation Amplitude (mm) | Residual Stiffness (N/mm) | Critical Stick–Slip Force (N) | Critical Stick–Lip Displacement (mm) | Power Exponent |
|---|---|---|---|---|
| 0.025 | 3270.9 | 266.11 | 0.0039 | −0.930 |
| 0.020 | 3280.7 | 266.78 | 0.0041 | −0.842 |
| 0.015 | 3290.1 | 266.66 | 0.0043 | −0.828 |
| 0.010 | 3297.2 | 265.82 | 0.0041 | −0.864 |
| 0.005 | 3318.6 | 265.35 | 0.0041 | −0.909 |

From Figure 14, it can be observed that the external excitation amplitude has minimal impact on the normalized four parameters of the Iwan model. The deviation of the four parameter values is generally within 10%. Increasing the external excitation amplitude has a negligible influence on these parameters. As a result, it can be deduced that the increasing of external excitation amplitude exerts little effect on the hysteresis characteristics of bolted joints. It can be concluded that, under the specified bolt preloading force and friction coefficient, the results of the same model can be used to predict the hysteresis loops for varying external excitation amplitudes, which demonstrates the applicability of the proposed constitution.

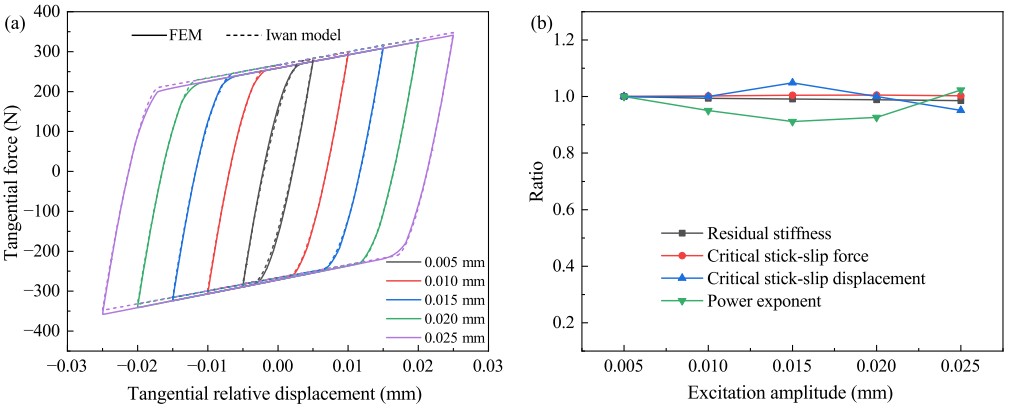

**Figure 14.** Modeling for bolted joint with different excitation amplitude: (**a**) Hysteresis nonlinearity and (**b**) Identified four parameters of Iwan model.

## 5. Conclusions

In this study, a memory effect penalty constitution was developed to characterize the hysteresis nonlinearity induced by the stick–slip friction contact behavior of joint interfaces. The proposed model more accurately represents the non-linear stiffness softening phenomenon on interfaces compared to the standard penalty method and the Lagrange multiplier method, resulting in simulation results that align more closely with theoretical predictions. Through FEA, the proposed constitution was used to simulate the hysteresis loops of a bolt joint and validated by comparison with the experimental results in the literature. This agreement confirmed the effectiveness of the proposed constitution and demonstrated a good performance of modeling the transition from the micro-stick contact regime to the macro-slip one.

The validated finite element method was then applied to investigate the effects of bolt clamping force, friction coefficient, and excitation amplitude on the hysteresis loops of the bolted joint. These simulated hysteresis loops were used to identify four parameters of the Iwan model. As the bolt preload and friction coefficient decrease, the critical stick–slip force and displacement nearly exhibit a linear decreasing trend, while the residual stiffness

indicates a slower decreasing trend. However, the impact of external excitation amplitudes on the four model parameters is slight.

**Author Contributions:** D.Y.: Methodology, Data curation, Software, Validation, Writing—original draft. D.W.: Conceptualization, Methodology, Writing—review and editing, Resources, Funding acquisition. Q.W.: Writing—review and editing, Supervision, Resources, Funding acquisition. All authors have read and agreed to the published version of the manuscript.

**Funding:** The work is supported by the National Natural Science Foundation of China (Nos. 52305141 and U2141212).

**Data Availability Statement:** Data will be made available on request.

**Conflicts of Interest:** The authors declare no conflicts of interest.

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
