# Peer review of "Modeling for Hysteresis Contact Behavior of Bolted Joint Interfaces with Memory Effect Penalty Constitution"

_machines, doi:10.3390/machines12030190_

Round 1

Reviewer 1 Report

Comments and Suggestions for Authors

The manuscript deals with the interesting topic of mapping the hysteresis behavior of bolted joints under periodic loading using the finite element method. It is written in an understandable way and the results are comprehensible. Nevertheless, there are some ambiguities and deficiencies, which are listed below and should be addressed:

· In Eqs. (10) and (11), F0 was introduced as the amplitude of the tangential stresses. In Eq.  (12), however, this amplitude was tacitly set equal to the maximum possible amplitude μp according to Coulombs law. However, cyclic tangential stresses can also have smaller amplitudes that do not yet cause complete sliding. So why do you take the critical value as the tangential stress amplitude? The same problem occurs on page 7. While Eqs. (19) and (20) still contain the tangential stress amplitude, the critical value is assigned to it in Eqs. (21) and (22).

·  On page 6 line 190, it is stated:

… By considering the isotropic of friction nature, the critical tangential stress is expressed as Equation (16) …

While Eqs. (14) and (15) as well as (18), (19) and (20) are reasonable solutions if the applied tangential loading is not directed exactly along one axis, I cannot understand the purpose of Eq. (16). What is your intention? The same applies to the incremental form Eq. (17). Is it used to be able to consider arbitrary loading histories?

· On page 7 in lines 204-206 it is mentioned that each element of the contact surface can be regarded as an asperity model and Eq. (22) is applied to map the memory-effect of the stick-slip contact. However, it should be noted that Mindlin only analyses one isolated single asperity contact. His theory does not consider the interaction of asperities, which is present in real contact problems. This interaction is neglected in your study, isn't it?

· On page 9 in lines 269-271 it s stated:

Five distinct contact pairs are established: bolt head-force washer, force washer-fixed specimen, fixed specimen-moving specimen, moving specimen-nut and bolt thread-nut thread from top to bottom

Has the novel memory-effect penalty constitution been applied to all 5 contact interfaces? If this is the case, how can the individual contributions of the contact interfaces to the total energy dissipation be deduced? The following hysteresis loops (e.g. from Figure 9 or 11) refer to the entire bolted joint. This does not guarantee that the individual contributions from the various contact interfaces are also suitably approximated. Please comment on this.

· Throughout the whole text the exponent of the monomial power function χ is called a "power series", even though there is no series at all, it is just an exponent. Please correct or explain it! Furthermore, the exponent in Eq. (24) is certainly not the same as that introduced in Eq. (9) and later set equal to 2.5. For this reason, a different designation should be used.

· On page 14 in lines 395-398 it is stated:

Moreover, the deterioration in bolt clamping performance also contributes to a decrease in the power series…

The term "series" certainly refers to the "exponent" again. However, the statement is not entirely correct. According to Table 2 and Figure 12(b), the exponent is smaller with a preload of 500N than with a preload of 300N. What is the reason for this?

· In Table 3, column 3:

The first critical stick-slip force for a friction coefficient μ=0.1 is specified as 890.03. This value must be wrong and corrected (too high by a factor of 10).

The full review is attached!

Reviewer 2 Report

Comments and Suggestions for Authors

The memory-effect penalty constitution was used to characterize the  hysteresis nonlinearity induced by the stick-slip friction contact behavior of a bolted  joint .

A simple numerical example with different models is studied and the comparison performed.The influence of many parameters are evaluated.

All the development and the results are very interesting. The paper can be accepted in the present form.

Comments on the Quality of English Language

 Minor editing of English language required

Reviewer 3 Report

Comments and Suggestions for Authors

1. Although the reviewer is not an English language expert, he recommends that the authors check the article with an expert. because there are some unfortunate terms in the article, e.g. Memory-effect penalty, This phenomenon can be attributed to the "elastic slip" effect

2. According to what criteria was the load on the joint determined by the axial force of the screw? The values quoted in the figure 9 are very small. 902N/ 20x20mm –πx7mm2/4= 2.48N/mm2. Please comment. What is the relationship with typical connection conditions for an M6 screw.

3. The content of Line 231-239 should be more prominent in the conclusions.

Round 2

Reviewer 1 Report

Comments and Suggestions for Authors

Dear authors,

Thank you for responding to all my comments in a satisfactory manner.

Kind regards!